# EHRDiff : Exploring Realistic EHR Synthesis with Diffusion Models

**Hongyi Yuan**[†]                                                                      *yuanhy20@mails.tsinghua.edu.cn*
*Center for Statistical Science*
*Tsinghua University*

**Songchi Zhou**[†]                                                                     *zhou-sc23@mails.tsinghua.edu.cn*
*Center for Statistical Science*
*Tsinghua University*

**Sheng Yu**[*]                                                                           *syu@tsinghua.edu.cn*
*Center for Statistical Science*
*Tsinghua University*

**Reviewed on OpenReview:** `https://openreview.net/forum?id=DIGkJhGeqi`

## Abstract

Electronic health records (EHR) contain a wealth of biomedical information, serving as valuable resources for the development of precision medicine systems. However, privacy concerns have resulted in limited access to high-quality and large-scale EHR data for researchers, impeding progress in methodological development. Recent research has delved into synthesizing realistic EHR data through generative modeling techniques, where a majority of proposed methods relied on generative adversarial networks (GAN) and their variants for EHR synthesis. Despite GAN-based methods attaining state-of-the-art performance in generating EHR data, these approaches are difficult to train and prone to mode collapse. Recently introduced in generative modeling, diffusion models have established cutting-edge performance in image generation, but their efficacy in EHR data synthesis remains largely unexplored. In this study, we investigate the potential of diffusion models for EHR data synthesis and introduce a novel method, EHRDIFF. Through extensive experiments, EHRDIFF establishes new state-of-the-art quality for synthetic EHR data, protecting private information in the meanwhile.

## 1 Introduction

Electronic health records (EHR) contain vast biomedical knowledge. EHR data may enable the development of state-of-the-art computational biomedical methods for dynamical treatment (Sonabend et al., 2020), differentiable diagnosis (Yuan & Yu, 2021), rare genetic disease identification (Alsentzer et al., 2022), etc. However, EHRs contain sensitive patients' private health information. Before being publicly accessible, real-world EHRs need to undergo de-identification (Johnson et al., 2016; 2023). The de-identification process uses automatic algorithms and requires tedious thorough human reviewing. Pending releasing approval can take months out of legal or ethical concerns (Hodge et al., 1999). Such circumstances limit the public release of rich EHR data, hence impeding the advancement of precision medicine methodologies. To mitigate the issue of limited publicly available EHR data, researchers alternatively explored generating synthetic EHR data (Choi et al., 2017; Walonoski et al., 2017). Realistic synthetic EHR generation has recently become a research field of medical informatics.

A line of work approached EHR data synthesis through generative modeling techniques, where they trained generative models on limited real EHR data to generate synthetic EHR data. Recent research developed

---

[*]Corresponding Author. [†] Contributed Equally. Codes are released in `https://github.com/sczzz3/EHRDiff.git`.

variants of auto-encoders (Vincent et al., 2008; Biswal et al., 2020) or generative adversarial networks (GAN) (Goodfellow et al., 2014; Choi et al., 2017). The majority of EHR data synthesis methodologies have relied on GAN (Choi et al., 2017; Baowaly et al., 2018; Zhang et al., 2019; Yan et al., 2020). Although GAN-based methods achieved state-of-the-art performance with respect to synthetic EHR quality and privacy preservation, they suffer from training instability and mode collapse (Che et al., 2017). Previous research proposed different techniques to mitigate the problem, while as shown in our experiments, GAN-based methods still are prone to such problems, resulting in unsatisfactory synthetic data quality. This may raise concerns when developing real-world systems using synthetic EHR data from GAN-based methods.

Most recently, novel diffusion models (Sohl-Dickstein et al., 2015) in generative modeling have been proposed and have achieved cutting-edge generation performance in the field of vision (Ho et al., 2020; Song et al., 2021), audio (Kong et al., 2021), or texts (Li et al., 2022; Gong et al., 2023; Yuan et al., 2022). Many variants of diffusion models have surpassed the generation performance of GANs in sample quality and diversity. In general, starting from random noise features, diffusion models use a trained denoising distribution to gradually remove noise from the features and ultimately generate realistic synthetic features. The efficacy of diffusion models on realistic EHR synthesis is less studied compared to GANs. Considering the superior performance of diffusion models in other domains, our work explores the synthesizing performance of such techniques on EHR data. We introduce EHRDIFF , a diffusion model-based EHR synthesizing model.

Our work conducts comprehensive experiments using publicly available real EHR data and compares the effectiveness of EHRDIFF against several other GAN-based EHR data synthesizing methods. We provide empirical evidence that EHRDIFF is capable of generating synthetic EHR data with a high degree of quality. Additionally, our findings reveal that the synthetic EHR data produced by EHRDIFF is of superior quality compared to those generated by GAN-based models, and it is more consistent with the distribution of real-world EHR data.

Our research has two primary contributions: Firstly, we introduce the use of diffusion models to the realm of realistic EHR synthesis and propose a diffusion-based method called EHRDIFF . Secondly, through extensive experimentation on publicly available EHR data, we demonstrate the superior quality of synthetic EHR data generated by EHRDIFF in comparison to GAN-based EHR synthesizing methods. Furthermore, the synthetic EHR data generated by EHRDIFF exhibits excellent correlation with real-world EHR data.

Our work is summarized as two following contributions:

1. We introduce diffusion models to the fields of EHR data synthesis and propose a diffusion-based method called EHRDIFF .

2. Through extensive experiments on publicly available real EHR data, we empirically demonstrate the superior generation quality of EHRDIFF over GAN-based EHR synthesis methods for various EHR feature formats, including categorical, continuous, and time-series features. In the meanwhile, EHRDIFF can safeguard private information in real training EHR.

## 2 Related Work

### 2.1 EHR data synthesis

In the literature on EHR synthesis, researchers are usually concerned with the generation of discrete code features such as ICD codes rather than clinical narratives. Researchers have developed various methods to generate synthetic EHR data. Early work was usually disease-specific or covered a limited number of diseases. Buczak et al. (2010) developed a method that generates EHR including visit records, clinical activity, and laboratory results for 203 synthetic tularemia outbreak patients of tularemia. The features in synthetic EHR data are generated based on retrieving similar real-world EHR which is inflexible and prone to privacy leakage. Walonoski et al. (2017) developed a software named Synthea which generates synthetic EHRs with various patient information based on publicly available data. They build generation workflows based on biomedical knowledge and real-world feature statistics. Various models are aggregated for different feature synthesis. However, Synthea only covered the 20 most common conditions.

Recently, researchers mainly applied generative modeling methods for EHR synthesis (Ghosheh et al., 2022). Medical GAN (medGAN) (Choi et al., 2017) introduced GAN to EHR synthesis. medGAN can generate synthetic EHR data with good quality and is free of tedious feature engineering. Following medGAN, various GAN-based methods are proposed, such as medBGAN (Baowaly et al., 2018), EHRWGAN (Zhang et al., 2019), CorGAN (Torfi & Fox, 2020a), etc. These GAN-based methods advance synthetic EHR to higher quality. However, a common drawback of GAN-based methods is that these methods suffer from the mode collapse phenomenon which results in a circumstance where a GAN-based model is capable of generating only a few modes of real data distribution (Thanh-Tung et al., 2018). To mitigate the problem, GAN methods for EHR generation rely on pre-trained auto-encoders to reduce the feature dimensions for training stability. However, inappropriate hyper-parameter choices and autoencoder pre-training will lead to sharp degradation of synthetic EHR quality or even failure to generate realistic data. There is also research that uses GAN-based models for conditional synthetic EHR generation to model the temporal structure of real EHR data (Zhang et al., 2020a). Since diffusion models are less studied in EHR synthesis, we focus on the unconditional generation of EHR and leave modeling conditional temporal structure with diffusion models to future work.

Besides GAN-based methods, there also exists research that explores generating synthetic EHR data through variational auto-encoders (Biswal et al., 2020) or language models (Wang & Sun, 2022). Concurrently, MedDiff (He et al., 2023) is proposed and explores diffusion models for synthetic EHR generation, and they propose a new sampling technique without which the diffusion model fails to generate high-quality EHRs. Ceritli et al. (2023) directly apply TabDDPM (Kotelnikov et al., 2022) to synthesizing EHR data.

## 2.2 Diffusion Models

Diffusion models are formulated with forward and reverse processes. The forward process corrupts real-world data by gradually injecting noise, and harvesting training data with different noise levels for a denoising distribution, while the reverse process generates realistic data by removing noise using the denoising distribution. Sohl-Dickstein et al. (2015) first proposed and provided theoretical support for diffusion models. Denoising diffusion probabilistic models (DDPM) (Ho et al., 2020) and noise conditional score networks (NCSN) (Song et al., 2021) have discovered the superior capability in image generation, and diffusion models have become a focused research direction since then. Recent research generalizes diffusion models to the synthesis of other data modalities and achieves excellent performance (Li et al., 2022; Kong et al., 2021). Our work is among the first research that introduces diffusion models to realistic EHR synthesis (He et al., 2023; Ceritli et al., 2023).

## 3 Method

In this section, we give an introduction to the problem formulation of realistic EHR synthesis and the technical details of our proposed EHRDIFF .

## 3.1 Problem Formulation

Real-world EHR contains diverse patient information including demographics, physiological conditions, laboratory results, ICD codes, etc. Such information may be presented in various data formats (e.g., categorical, binary, continuous, etc). Features of different information may also have different scales, such as ages and heights. In this work, we use a vector $\boldsymbol{x}_0$ to represent the EHR, where $\boldsymbol{x}_0 \in \mathcal{R}^{|\mathcal{C}|}$ and $\mathcal{C}$ represents the dimension of EHR features of interest. We treat all features of different formats as real numbers ranging from 0 to 1. Following previous research (Choi et al., 2017; Baowaly et al., 2018) which only focused on binary code data, we use 1 standing for code occurrence and 0 otherwise. For continuous and categorical features, we normalize each feature within the range.

## 3.2 General Framework for Diffusion Model

Diffusion models are characterized by forward and reverse Markov processes with latent variables where the forward process transforms the real samples to random Gaussian noise while the reverse process generates

synthetic samples by gradual denoising the noise. As demonstrated by Song et al. (2021), the forward and reverse processes can be described by stochastic differential equations (SDE). The general SDE form for modeling the forward process follows:

$$d\boldsymbol{x} = f(\boldsymbol{x}, t)dt + g(t)d\boldsymbol{w}, \tag{1}$$

where $\boldsymbol{x}$ represents data points, $\boldsymbol{w}$ represents the standard Wiener process, $t$ is diffusion time, and ranges from 0 to $T$. At $t = 0$, $\boldsymbol{x}_0$ follows real data distribution while at $t = T$, $\boldsymbol{x}_T$ asymptotically follows a random Gaussian distribution. Functions $f$ and $g$ respectively define the sample corruption pattern and the level of injected noises. $f$ and $g$ together corrupt real-world samples to random noise. Based on the forward SDE, the SDE for the reverse process can be derived as follows:

$$d\boldsymbol{x} = \left(f(\boldsymbol{x}, t) - g^2(t)\nabla_{\boldsymbol{x}} \log p_t(\boldsymbol{x})\right) dt + g(t)d\boldsymbol{w}, \tag{2}$$

where $p_t(\boldsymbol{x})$ is the marginal density of $\boldsymbol{x}$ at time $t$, and $\nabla_{\boldsymbol{x}} \log p_t(\boldsymbol{x})$ is the score function, which indicates a vector field of which the direction is pointed to the high-density data area. With reparameterization proposed in Karras et al. (2022), the reverse generation process can also be described with the probability flow ordinary differential equations (ODE) instead of SDEs (Song et al., 2021):

$$d\boldsymbol{x} = -\dot{\sigma}(t)\sigma(t)\nabla_x \log p_t\big(\boldsymbol{x}; \sigma(t)\big)dt = -\dot{\sigma}(t)\sigma(t)\nabla_x \log p_{\sigma_t}\big(\boldsymbol{x}\big)dt, \tag{3}$$

$$h(t) = \exp\left(\int_0^t f(\xi)d\xi\right), \tag{4}$$

$$\sigma(t) = \sigma_t = \sqrt{\int_0^t \frac{g(\xi)^2}{h(\xi)^2}d\xi}, \tag{5}$$

where $\dot{\sigma}(t)$ represents the derivative of $\sigma(t)$. SDE and probability flow ODE indicate stochastic and deterministic generation processes respectively. To generate synthetic samples following reverse processes, it is required to learn a score function $s_\theta(\boldsymbol{x})$ parameterized by $\theta$ by score matching $\min_\theta \mathbb{E}_{p(\boldsymbol{x}_0)p_{\sigma_t}(\boldsymbol{x}|\boldsymbol{x}_0)} \left[\|s_\theta(\boldsymbol{x}) - \nabla_{\boldsymbol{x}} \log p_{\sigma_t}(\boldsymbol{x}|\boldsymbol{x}_0)\|_2^2\right]$.

### 3.3 EHRDiff

For the diffusion process of EHRDIFF , we use $h(t) = 1$ and $\sigma_t = t$ from previous research (Karras et al., 2022). Therefore $\nabla_{\boldsymbol{x}} \log p_{\sigma_t}(\boldsymbol{x}|\boldsymbol{x_0}) = -\frac{\boldsymbol{x}-\boldsymbol{x_0}}{\sigma_t^2}$ and we reparameterize $s_\theta(\boldsymbol{x}) = -\frac{\boldsymbol{x}-D_\theta(\boldsymbol{x},\sigma_t)}{\sigma_t^2}$, then the objective can be derived as:

$$\min_\theta \mathbb{E}_{p(\boldsymbol{x}_0)p_{\sigma_t}(\boldsymbol{x}|\boldsymbol{x}_0)} \left[\left\|\frac{D_\theta(\boldsymbol{x}, \sigma_t) - \boldsymbol{x}_0}{\sigma_t^2}\right\|_2^2\right], \tag{6}$$

and with further simplification of ignoring $\sigma_t^2$, the final objective becomes:

$$\min_\theta \mathbb{E}_{p(\boldsymbol{x}_0)p_{\sigma_t}(\boldsymbol{x}|\boldsymbol{x}_0)} \left[\|D_\theta(\boldsymbol{x}, \sigma_t) - \boldsymbol{x}_0\|_2^2\right]. \tag{7}$$

Generally, $D_\theta(\boldsymbol{x}, \sigma_t)$ can be modeled by neural networks, while such direct modeling may cause obstacles for optimization because the variance of $\boldsymbol{x}_t$ and the scale of $\sigma_t$ are diverse at different time step $t$. Therefore, we chose the pre-conditioning design of $D_\theta(\boldsymbol{x}, \sigma_t)$ (Karras et al., 2022) where $D_\theta(\boldsymbol{x}, \sigma_t)$ is decomposed as:

$$D_\theta(\boldsymbol{x}; \sigma) = c_{\text{skip}}(\sigma)\boldsymbol{x} + c_{\text{out}}(\sigma)F_\theta(c_{\text{in}}(\sigma)\boldsymbol{x}; c_{\text{noise}}(\sigma)). \tag{8}$$

$F_\theta$ is modeled with neural networks and such designs of $c_{\text{in}}$ and $c_{\text{noise}}$ regulate the input to the network to be unit variance across different time step $t$, and $c_{\text{out}}$ and $c_{\text{skip}}$ together set the neural model prediction to be unit variance with minimized scale. Therefore in EHRDIFF , we chose $c_{\text{out}} = \sigma\sigma_{\text{data}}/\sqrt{\sigma^2 + \sigma_{\text{data}}^2}$ and $c_{\text{skip}}(\sigma) = \sigma_{\text{data}}^2/(\sigma^2 + \sigma_{\text{data}}^2)$. $c_{\text{noise}}(\sigma) = 0.25 \ln \sigma$ which is designed empirically with the principle of constraining the input noise scale from varying immensely and $\ln(\sigma) \sim \mathcal{N}(P_{\text{mean}}, P_{\text{std}}^2)$ following Karras et al. (2022), where $P_{\text{mean}}$ and $P_{\text{std}}$ are hyper-parameters to be set.

With the aforementioned formalization of $h(t)$, $\sigma(t)$ and the learned score function $s_\theta$, the generation process of EHRDIFF can be expressed as the ODE:

$$\mathrm{d}\boldsymbol{x} = -ts_\theta(\boldsymbol{x})\mathrm{d}t. \tag{9}$$

Solving the ODE numerically requires discretization of the time step $t$ and a proper design of noise level $\sigma_t$ along the solution trajectory. Therefore, following previous research (Karras et al., 2022), we set the maximum and minimum noise levels as $\sigma_{\max}$ and $\sigma_{\min}$, and use the following form of discretization:

$$t_i = \sigma_{t_i} = \left( (\sigma_{\max})^{\frac{1}{\rho}} + \frac{i}{N-1} \left( (\sigma_{\min})^{\frac{1}{\rho}} - (\sigma_{\max})^{\frac{1}{\rho}} \right) \right)^{\rho}, \tag{10}$$

where $i$'s are integers and range from 0 to $N$, $\sigma_{t_N} = 0, \sigma_{t_{N-1}} = \sigma_{\min}$, and $\rho$ controls the schedules of discretized time step $t_i$ and trades off the discretized strides $t_i - t_{i-1}$ the larger value of which indicates a larger stride near $t_0$. In order to solve the ODE more precisely and generate synthetic EHR with higher quality, we use Heun's 2nd order method, which adds a correction updating step for each $t_i$ and alleviates the truncation errors compared to the 1st order Euler method. We leave the detailed sampling procedure to Appendix A.

## 4 Experiments

To demonstrate the effectiveness of our proposed EHRDIFF , we conduct extensive experiments evaluating the quality of synthetic EHRs and the privacy concerns of the method. We also compare EHRDIFF the several GAN-based realistic EHR synthesis methods to illustrate the performance of EHRDIFF .

### 4.1 Dataset

Many previous research uses in-house EHR data which is not publicly available for method evaluation (Zhang et al., 2019; Yan et al., 2020). Such experiment designs set obstacles for later research to reproduce experiments. In this work, we use a publicly available EHR database, MIMIC-III, to evaluate EHRDIFF .

Deidentified and comprehensive clinical EHR data is integrated into MIMIC-III (Johnson et al., 2016). The patients are admitted to the intensive care units of the Beth Israel Deaconess Medical Center in Boston. For each patient's EHR for one admission, we extract the diagnosis and procedure ICD-9 code and truncate the ICD-9 code to the first three digits. This preprocessing can reduce the long-tailed distribution of the ICD-9 code distribution and results in a 1,782 code set. Therefore, the EHR for each patient is formulated as a binary vector of 1,782 dimensions. The final extracted number of EHRs is 46,520 and we randomly select 41,868 for model training while the rest are held out for evaluation.

Although most of the existing research focused on synthesizing discrete code features, real-world EHR data contains various data formats such as continuous test results values or time series of electrocardiograms (ECG). In this work, we extend the previous research and explore applying EHRDIFF to the synthesis of EHR data other than binary codes. We use the following two datasets: CinC2012 Data and PTB-ECG Data. CinC2012 Data is a dataset for predicting the mortality of ICU patients and contains various feature formats such as categorical age, or continuous serum glucose values. PTB-ECG Data contains ECG signal data for heart disease diagnosis. Detailed introductions of both datasets are left in C. All the categorical features are converted into binary columns by one-hot encoding, and continuous features are normalized to values in the range of $[0.0, 1.0]$. We use sets A and B in CinC2023 Data as training and held-out testing sets respectively. The PTB-ECG Data is split with a ratio of 8:2 for training and held-out testing.

### 4.2 Baselines

To better demonstrate EHR synthesis performance, we compare EHRDIFF to several strong baseline models as follows.

**medGAN** (Choi et al., 2017) is the first work that introduces GAN to generating realistic synthetic EHR data. Considering the obstacle of directly using GAN to generate high-dimensional binary EHR

vectors, medGAN alters to a low-dimensional dense space for generation by taking advantage of pre-trained auto-encoders. The model generates a dense EHR vector and then recovers a synthetic EHR with decoders.

**medBGAN and medWGAN** (Baowaly et al., 2018) are two improved GAN models for realistic EHR synthesis. medGAN is based on the conventional GAN model for EHR synthesis, and such a model is prone to mode collapse where GAN models may fail to learn the distribution of real-world data. medBGAN and medWGAN integrate Boundary-seeking GAN (BGAN) (Hjelm et al., 2018) and Wasserstein GAN (WGAN) (Adler & Lunz, 2018) respectively to improve the performance of medGAN and stabilize model training.

**CorGAN** (Torfi & Fox, 2020b) is a novel work that utilizes convolutional neural networks (CNN) instead of multilayer perceptrons (MLP) to model EHR data. Specifically, they use CNN to model the autoencoder and the generative network. They empirically elucidate through experiments that CNN can perform better than the MLP in this task.

**EMR-WGAN** (Zhang et al., 2020b) is proposed to further refine the GAN models from several perspectives. To avoid model collapse, the authors take advantage of WGAN. The most prominent feature of EMR-WGAN is that it is directly trained on the discrete EHR data, while the previous research universally uses an autoencoder to first transform the raw EHR data into low-dimensional dense space. They utilize BatchNorm (Ioffe & Szegedy, 2015) for the generator and LayerNorm (Ba et al., 2016) for the discriminator to improve performance. As is shown in their experiments, these modifications significantly improve the performance of GAN.

### 4.3 Evaluation Metric

In our experiments, we evaluate the generative models' performance from two perspectives: utility and privacy (Yan et al., 2022). Utility metrics evaluate the quality of synthetic EHRs and privacy metrics assess the risk of privacy breaches. In the following metrics, we generate and use the same number of synthetic EHR samples as the number of real training EHR samples.

#### 4.3.1 Utility Metrics

We follow previous research for a set of utility metrics. The following metrics evaluate synthetic EHR quality from diverse perspectives.

**Dimension-wise distribution** describes the feature-level resemblance between the synthetic data and the real data. The metric is widely used in previous research to investigate whether the generative model is capable of learning the high-dimensional distribution of real EHR data. For each code dimension, we calculate the empirical mean estimation for synthetic and real EHR data respectively. The mean estimation indicates the prevalence of the code. We visualize the dimension-wise distribution using scatter plots where both axes represent the prevalence of synthetic and real EHR respectively. Many codes have very low prevalence in real EHR data. The generation model may be prone to mode collapse and fail to generate the codes with low prevalence. Therefore, we count the number of codes that exist in the synthetic EHR samples and dub the quantity non-zero code columns.

**Dimension-wise correlation** measures the difference between the feature correlation matrices of real and synthetic EHR data. The $i, j$ entry of correlation matrices calculates the Pearson correlation between the $i$th and $j$th features. For both the synthetic and real EHR data, we calculate first the correlation matrices, and then the averaged absolute differences between the correlation matrices. We name this metric the correlation matrix distance (CMD).

**Dimension-wise prediction** evaluates whether generative models capture the inherent code feature relation by designing classification tasks. Specifically, we select one of the code features to be the classification target and use the rest of the features as predictors. To harvest a balanced target distribution, we sort the code features according to the entropy $H(p)$ of code prevalence $p$, where $H(p) = -p \log(p) - (1-p) \log(1-p)$.

Table 1: NZC represents Non-Zero code Columns, CMD represents Correlation Matrix Distance. ↓ and ↑ indicate the respectively lower and higher numbers for better results.

|  | NZC ($\uparrow$) | CMD ($\downarrow$) |
|---|---|---|
| medGAN | 643±59.4 | 45.652±11.911 |
| medBGAN | 898±36.8 | 63.186±16.359 |
| medWGAN | 376±33.5 | 8.603±0.163 |
| CorGAN | 753±125.1 | 10.997±0.420 |
| EMR-WGAN | 1060±29.6 | 8.173±0.274 |
| EHRDIFF | **1770**±1.9 | **7.769**±0.013 |

We select the top 30 code features according to entropies and form 30 individual classification tasks. For each task, we fit a classification model with logistic regression using real training and synthetic EHR data and assess the F1 score on the preset evaluation real EHR data.

### 4.3.2 Privacy Metrics

Generative modeling methods need real EHR data for training which raises privacy concerns among practitioners. Attackers may infer sensitive private information from trained models. Besides the utility of synthetic EHR data, we also evaluate existing models from a privacy protection perspective (Choi et al., 2017; Zhang et al., 2019; Yan et al., 2022).

**Attribute inference risk**   describes the risk that sensitive private information of real EHR training data may be exposed based on the synthetic EHR data It assumes a situation where the attackers already have several real EHR training samples with partially known features, and try to infer the rest features through synthetic data. Specifically, we assume that attackers first use the k-nearest neighbors method to find the top $k$ most similar synthetic EHRs to each real EHR based on the known code features, and then recover the rest of unknown code features by majority voting of $k$ similar synthetic EHRs. We set $k$ to 1 and use the most frequent 256 codes as the features known by the attackers. The metric is quantified by the prediction F1-score of the unknown code features.

**Membership inference risk**   evaluates the risk that given a set of real EHR samples, attackers may infer the samples used for training based on synthetic EHR data. We mix a subset of training real EHR data and held-out testing real EHR data to form an EHR set. For each EHR in this set, we calculate the minimum L2 distance with respect to the synthetic EHR data. The EHR whose distance is smaller than a preset threshold is predicted as the training EHR. We report the prediction F1 score to demonstrate the performance of each model under membership inference risk.

### 4.4 Implementation Detail

In our experiments, for the diffusion noise schedule, we set $\sigma_{\min}$ and $\sigma_{\max}$ to be 0.02 and 80. $\rho$ is set to 7 and the time step is discretized to $N = 32$. $P_{mean}$ is set to $-1.2$ and $P_{std}$ is set to 1.2 for noise distribution in the training process. For $F_\theta$ in Equation 8, it is parameterized by an MLP with ReLU (Nair & Hinton, 2010) activations and the hidden states are set to $[1024, 384, 384, 384, 1024]$. For the baseline methods, we follow the settings reported in their papers. The reported standard errors marked with $\pm$ are calculated under 5 different runs.

### 4.5 Results on MIMIC

### 4.5.1 Utility Results

Figure 1 depicts the dimension-wise prevalence distribution of synthetic EHR data against real data. The scatters from EHRDIFF are distributed more closely to the diagonal dashed line compared to other baseline

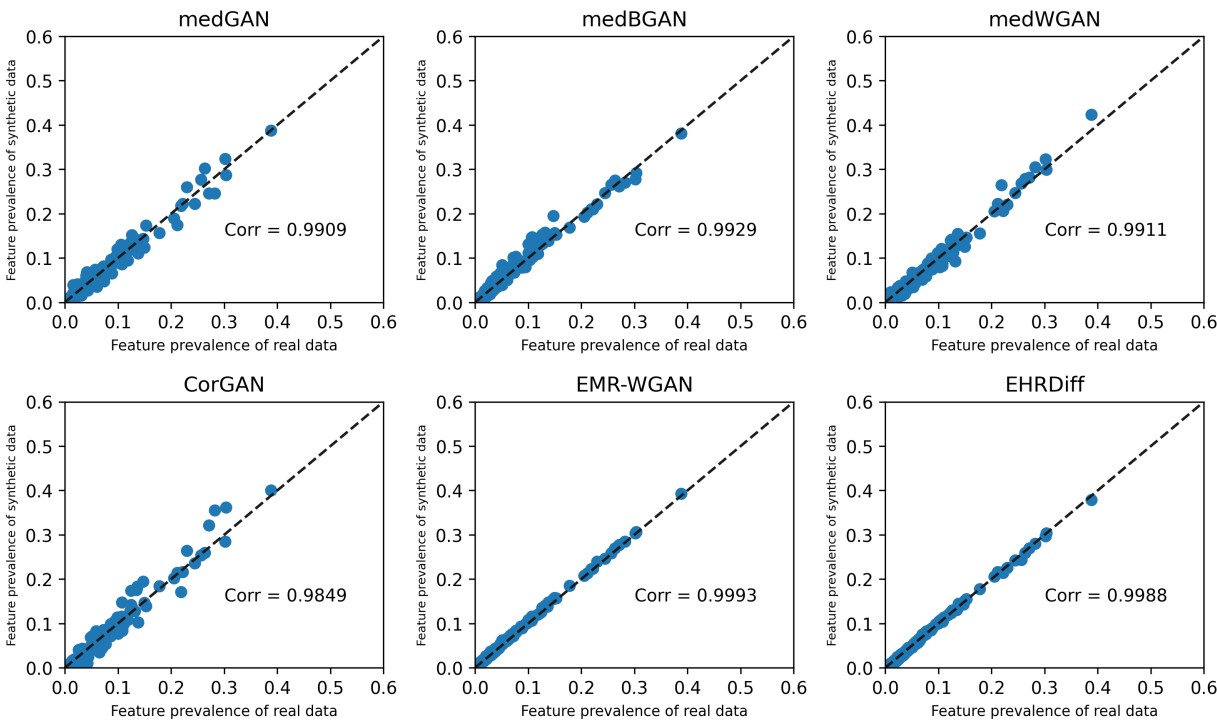

Figure 1: The dimension-wise probability scatter plot of synthetic EHR data from different generative models against real EHR data. The diagonal lines represent the perfect match of code prevalence between synthetic and real EHR data.

models, and EHRDIFF and EMR-WGAN achieve near-perfect correlation. As shown in Table 1, EHRDIFF outperforms all baseline methods in non-zero code column number (NZC) by large margins. This shows that GAN-based baselines all suffer from model collapse to different extents. The GAN-based method of best performance, EMR-WGAN, still fails to generate 722 code features with the same number of synthetic EHR samples as the real data. Although EMR-WGAN achieves a near-perfect correlation between real and synthetic code prevalence and is slightly better than EHRDIFF , NZC demonstrates that the correlation can be biased by high prevalence features and overshadow the evaluation of low prevalence features. The results above demonstrate that EHRDIFF can better capture the code feature prevalence of the real data than the GAN-based baselines, and is free from mode collapse. The synthetic EHR data by EHRDIFF has better diversity than that by GAN-based methods.

From CMD results in Table 1, EHRDIFF surpasses all baseline models. CMD results demonstrate that EHRDIFF can better capture the inherent pair-wise relations between code features than GAN-based methods. The F1 score scatters in Figure 2 of EHRDIFF are closer to the diagonal lines and achieve the highest correlation value as compared to baselines. This means that training on synthetic EHR data by EHRDIFF can lead to more similar performance to training on real data, and demonstrates that EHRDIFF can better model complex interactions between code features than baselines. It is indicated that synthetic EHR data by EHRDIFF may have superior utility for training downstream models biomedical tasks. We present more results on other utility metrics in B.

### 4.5.2 Privacy Results

In Table 2, we list the results against privacy attacks. In terms of attribute inference risk and membership inference risk, EHRDIFF achieves intermediate results, while medGAN and CorGAN respectively achieve the best results on attribute inference risk and membership inference risk. However, as shown in utility results, the quality of synthetic EHR data by both models is far worse than EHRDIFF . In an extreme circumstance

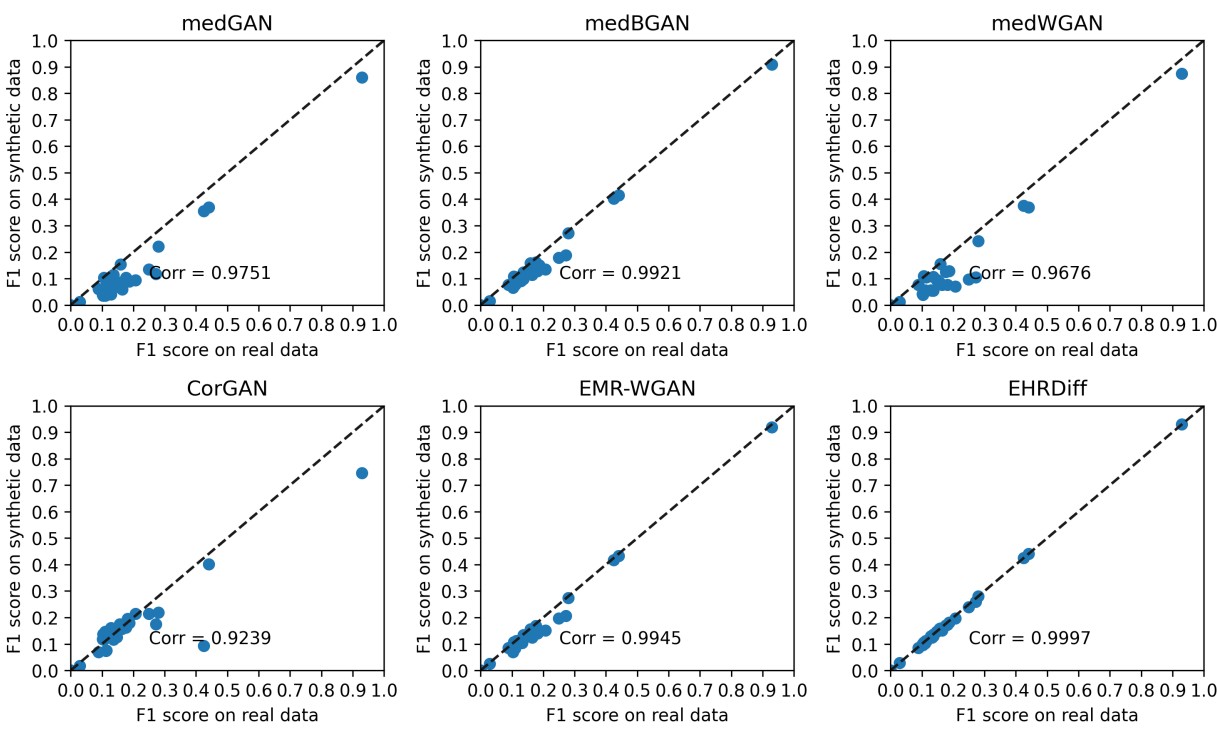

Figure 2: The dimension-wise prediction scatter plot of synthetic EHR data from different generative models against real EHR data. The diagonal lines represent the perfect match of code prediction between synthetic and real EHR data. Each scatter represents a task.

Table 2: The privacy assessment for each model. ↓ indicates the lower numbers for better results.

|  | Attribute Inference Risk (↓) | Membership Inference Risk (↓) |
| --- | --- | --- |
| medGAN | **0.0011**±0.0006 | 0.2941±0.0061 |
| medBGAN | 0.0069±0.0025 | 0.2961±0.0047 |
| medWGAN | 0.0066±0.0014 | 0.2928±0.0055 |
| CorGAN | 0.0033±0.0025 | **0.2413**±0.0061 |
| EMR-WGAN | 0.0259±0.0008 | 0.2943±0.0028 |
| EHRDIFF | 0.0190±0.0013 | 0.2956±0.0013 |

where a generative model fails to fit the real EHR data distribution, the model may achieve perfect results on both privacy metrics, since attackers can not infer private information through synthetic data of bad quality. Therefore, there exists an implicit trade-off between utility and privacy. We suspect that medGAN and CorGAN can better safeguard privacy due to mediocre synthesis quality. When compared to EMR-WGAN which achieves the best synthesis quality among baselines, EHRDIFF surpasses EMR-WGAN on attribute inference risk and achieves on-par results in terms of membership inference risk. To conclude, EHRDIFF can well protect the sensitive private information of real EHR training data.

### 4.6 Results on PTB-ECG and CinC2023

Since both datasets are designed for classification, we inspect the utilities of synthesized data by evaluating the Area Under receiver operating characteristic Curve (AUC) of classifiers trained with synthetic data. We

Table 3: The AUC values on Cinc2012 Data and PTB-ECG Data.

|  | Cinc2012 Data | PTB-ECG Data |
|---|---|---|
| Real | 0.8479 | 0.9963 |
| medGAN | 0.6176±0.0676 | 0.7550±0.0375 |
| medBGAN | 0.5942±0.0763 | 0.7301±0.0215 |
| medWGAN | 0.7012±0.0471 | 0.8071±0.0273 |
| CorGAN | 0.6352±0.1259 | 0.4521±0.0709 |
| EMR-WGAN | 0.8010±0.0143 | 0.8011±0.0171 |
| EHRDiff | **0.8405**±0.0024 | **0.9898**±0.0010 |

Table 4: Ablation results of utility and privacy on different aspects of EHRDIFF designs. VE and VP are short for variance exploding and variance preserving respectively. Pre-Cond represents pre-conditioning.

|  | Utility | | | Privacy | |
|---|---|---|---|---|---|
|  | Corr (↑) | NZC(↑) | CMD (↓) | AIR (↓) | MIR (↓) |
| EHRDIFF | 0.9989±0.0001 | 1771.0±1.4 | 7.768±0.013 | 0.0187±0.0010 | 0.2954±0.0014 |
| **on Diffusion Process** | | | | | |
| w/ VE | 0.9876±0.0003 | 1553.0±5.0 | 8.057±0.022 | 0.0130±0.0010 | 0.2549±0.0023 |
| w/ VP | 0.9976±0.0001 | 1564.3±1.2 | 8.002±0.013 | 0.0143±0.0007 | 0.3024±0.0007 |
| **on Sampling Method** | | | | | |
| w/ 1st Euler | 0.9974±0.0001 | 1541.3±4.2 | 7.889±0.006 | 0.0155±0.0003 | 0.3111±0.0012 |
| **on Network Design** | | | | | |
| w/o Pre-Cond | 0.9655±0.0027 | 432.0±26.8 | 8.719±0.058 | 0.0030±0.0009 | 0.3184±0.0018 |

use LightGBM (Ke et al., 2017) as classifiers and train on synthetic data of the same size as real training data.

The results shown in Table 3 that classifiers trained by synthetic data from EHRDIFF achieve the highest AUC values and are consistently better than GAN-based methods, reaching 0.8405 and 0.9898 on average on CinC2012 Data and PTB-ECG Data respectively. They also have on-par performance with classifiers trained by real data. The results show the great utility of EHRDIFF generated EHR data, and the efficacy is consistently good across different EHR data feature formats. This demonstrates EHRDIFF is practical in real-world scenarios and can approach EHR synthesis of diverse formats. The downstream biomedical models can benefit from training on synthetic EHR data by EHRDIFF, potentially overcoming the obstacles of limited publicly available real EHR data.

## 5 Discussion

### 5.1 Ablation Study

In this section, we discuss and conduct ablation experiments on the effective designs of EHRDIFF. The designs of EHRDIFF include three major aspects: diffusion process, sampling method for inference, and neural network design for modeling denoising functions. We demonstrate the performance for each through the utility and privacy metrics on MIMIC-III. The results are shown in Table 4. For the diffusion process, we compare the diffusion process of EHRDIFF to the choice of variance exploding (VE) and variance preserving (VP) diffusion process. VE and VP are diffusion processes originally proposed in Song et al. (2021). We can see that EHRDIFF outperforms both choices in terms of utility metrics while regarding the privacy concerns, EHRDIFF performs a little inferior to VP and VE diffusion processes. As discussed in Section 4.5.2, the privacy performance can be compromised to better utility performance. For the sampling process, Heun's 2nd order method outperforms Euler's 1st method in utility. A 2nd order sampling method can lead to a better step-wise estimation of each denoising step, and hence has better generation performance than 1st methods

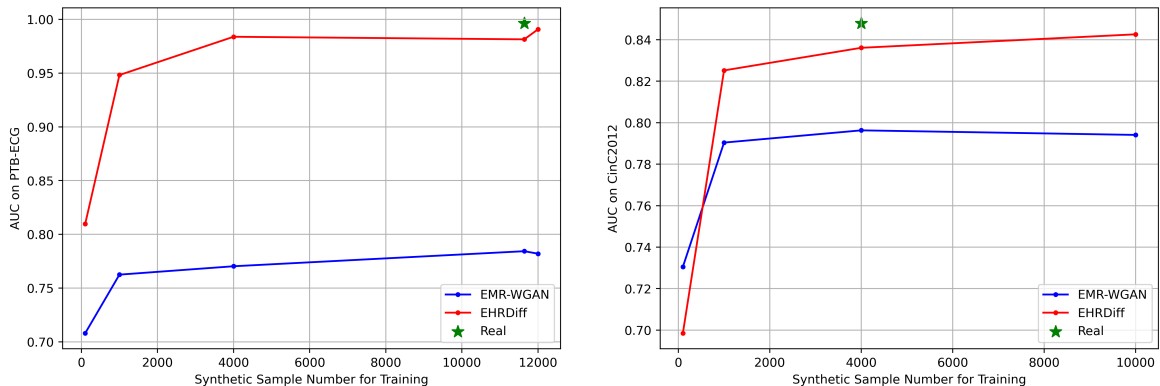

Figure 3: The line plots for CinC2012 and PTB-ECG with different data scales. The green star represents the performance of the model trained on real data.

when using the same trained denoising function. For the network design, we can see that EHRDIFF without pre-conditioning fails to generate high-quality EHR data as shown by the lowest NZC results of only 432. The possible explanation for this is that the binary code features may have diverse prevalence which leads to much difference in variance of each feature. Pre-conditioning sets the inputs and outputs of the neural networks to be unit variance and thus eases the modeling difficulty for the network in the denoising function.

Overall, we have demonstrated the effectiveness of the design choices in EHRDIFF regarding diffusion processes, sampling methods, and network designs. EHRDIFF achieves the best performance regarding utility metrics, indicating a superior generation quality, while such designs in EHRDIFF only compromise the privacy concerns marginally.

## 5.2 Influence of Synthetic Data Scale

Our main motivation is to use the synthetic data from EHRDIFF to aid the downstream methodology development on EHR, emphasizing the limitation of scarce real-world EHR data. Therefore, we further demonstrate the influence of synthetic EHR data scales on training downstream models. We train classifiers of the same type on PTB-ECG and CinC2023 data with different scales of synthetic data from EHRDIFF and the previous state-of-the-art EMR-WGAN for comparison. The results are shown in Figure 3.

As can be seen from the results, the downstream performance of models trained on synthetic data from EHRDIFF improves as the size of synthesized data scales up. Models trained on synthetic data generated from EHRDIFF outperformed those trained on data from EMR-WGAN across different sizes of synthetic training data for both CinC2012 and PTB-ECG, except on minimal data size of 100 on CinC2012. Notably, our results demonstrate that smaller sample sizes of synthetic EHR data from EHRDIFF can lead to superior downstream model performance compared to larger sample sizes from EMR-WGAN.

## 6 Potential Social Impacts

The generative models including our EHRDIFF for synthesizing EHR data can potentially reveal their training data, resulting in unexpected patient privacy information leakage. Although in our experiments we conduct extensive experiments for privacy protection from attacking risks following existing research (Yan et al., 2022), we also find the potential drawbacks of the privacy assessment metrics from our results and the extent to which applying our generative model in real-world application require further studying.

Data-driven generative models are prone to generating content with potential biases in training data. The EHR synthetic models may also suffer from such malicious tendencies of gender or demographic bias. Existing research on generative models has proposed methods to measure (Teo & Cheung, 2021) or mitigate such

problem (Grover et al., 2019b;a; Teo et al., 2022). Assessing and alleviating the bias of generative models for EHR synthesis remains an important research question.

## 7 Conclusion

In this work, we explore EHR data synthesis with diffusion models. We proposed EHRDIFF , a diffusion-based model, for EHR data synthesis. Through comprehensive experiments on binary code EHR data, we empirically demonstrate the superior performance in generating high-quality synthetic EHR data from multiple evaluation perspectives, setting new state-of-the-art EHR synthesis methods. In the meanwhile, we also show EHRDIFF can safeguard sensitive private information in real EHR training data. Furthermore, beyond binary code features in EHR data, the efficacy of EHRDIFF consistently excels in continuous and time-series features. EHRDIFF can help downstream biomedical methodology research overcome the obstacles of limited publicly available real EHR data.

**Acknowledgment**

This work was supported by the Natural Science Foundation of China (Grant No. 12171270) and the Natural Science Foundation of Beijing Municipality (Grant No. Z190024).

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

---

**Algorithm 1** Heun's 2nd Method for Sampling

---

**Input:** Time Step $t_i$ and noise level $\sigma_{t_i}$

1: Using Equation 9, calculate the derivative $\boldsymbol{g}_{t_i} = \mathrm{d}\boldsymbol{x}/\mathrm{d}t$:
$\boldsymbol{g}_{t_i} = t_i^{-1}\boldsymbol{x}_{t_i} - t_i^{-1}D(\boldsymbol{x}_{t_i};\sigma_{t_i})$,

2: Get intermediate $\tilde{\boldsymbol{x}}_{t_{i+1}}$ by taking Euler step:
$\tilde{\boldsymbol{x}}_{t_{i+1}} = \boldsymbol{g}_{t_i}(t_{i+1} - t_i) + \boldsymbol{x}_{t_i}$,

3: Calculate the gradient correction $\tilde{\boldsymbol{g}}_{t_i}$:
$\tilde{\boldsymbol{g}}_{t_i} = t_{i+1}^{-1}\tilde{\boldsymbol{x}}_{t_{i+1}} - t_{i+1}^{-1}D(\tilde{\boldsymbol{x}}_{t_{i+1}};\sigma_{t_{i+1}})$,

4: Get next time step sample $\boldsymbol{x}_{t_{i+1}}$:
$\boldsymbol{x}_{i+1} = \boldsymbol{x}_i + (t_{i+1} - t_i)(\frac{1}{2}\boldsymbol{g}_i + \frac{1}{2}\tilde{\boldsymbol{g}}_{i+1})$

5: **return** $\boldsymbol{x}_{t_{i+1}}$

---

Table 5: MCAD represents Medical Concept Abundance Distance. ↓ and ↑ indicate the respectively lower and higher numbers for better results.

|  | Latent Distance (↓) | MCAD (↓) |
|---|---|---|
| medGAN | -4.300±0.009 | 0.257±0.014 |
| medBGAN | -5.407±2.452 | 0.123±0.010 |
| medWGAN | -12.968±1.600 | 0.076±0.024 |
| CorGAN | -9.917±0.771 | 0.129±0.053 |
| EMR-WGAN | -14.437±0.549 | 0.101±0.008 |
| EHRDIFF | -13.560±0.211 | 0.076±0.002 |

## A  Sampling Algorithm

## B  Additional Results

The quality of synthetic EHR data can be evaluated from a multifaceted perspective (Yan et al., 2022). We use additional metrics to further evaluate the synthetic EHR data on MIMIC-III.

### B.1  Latent cluster analysis

The metric evaluates the distributional difference between the synthetic and real EHR data in the latent space. The metric first uses principle component analysis to reduce the sample dimension for both data and then cluster the samples in the latent space. Ideally, if synthetic and real EHR data are identically distributed, the synthetic and real EHR samples should respectively comprise half of the samples in one cluster. Therefore, the metric is calculated as:

$$\log\left(\frac{1}{K}\sum_{i=1}^{K}[\frac{n_i^{\mathrm{real}}}{n_i} - 0.5]^2\right), \tag{11}$$

where $K$ is the number of resulted clusters, $n_i$ and $n_i^{\mathrm{real}}$ are the sample number and the real sample number in $i$th cluster, respectively. The lower the value, the less synthetic data distribution deviates from the real data distribution. In our experiments, $K$ is decided by the elbow method (Yuan & Yang, 2019) for each synthetic data and in this work is 4 or 5 according to different methods.

### B.2  Medical concept abundance

The metric assesses the synthetic EHR data distribution on the record level. The metric calculates the empirical distribution of the unique positive (occurred) code number within each sample. The empirical

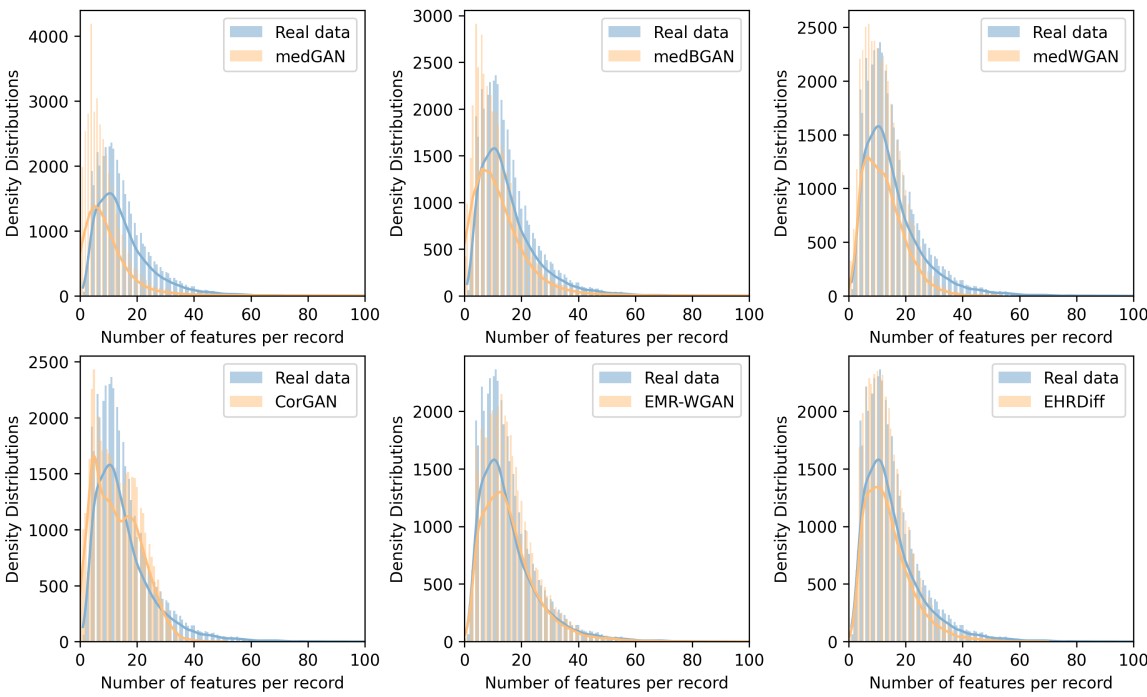

Figure 4: The histograms plot the empirical distributions of the unique code counts on the sample level. The solid lines are the kernel density estimations of the distribution.

distributions are calculated by histograms. The discrepancy between synthetic and real EHR data is calculated as follows:

$$\sum_{i=1}^{M} \frac{1}{2N} |h_r(i) - h_s(i)|, \tag{12}$$

where $M$ is the number of bins in histograms, $N$ denotes the number of samples for real (or synthetic) data, and $h_r(i)$ and $h_s(i)$ respectively represent the $i$th bin in the histograms of real and synthetic EHR data. In this work, M is set to 20.

### B.3 Results

From Table 5, it is shown that EHRDIFF performs better than most baselines and only marginally falls behind EMR-WGAN by 0.877 on the latent distance metric. In terms of MCAD, EHRDIFF consistently outperforms all baselines, and as depicted in Figure 4, we can see that the histogram of unique code count distribution of synthetic EHR data by EHRDIFF achieves the best fit to that of real EHR data. From a sample-level perspective, latent distance results illustrate that synthetic EHR data by EHRDIFF is closely distributed to real EHR data. The MCAD results show that synthetic data by EHRDIFF resembles the real EHR most in terms of unique positive code counts. This result is in line with the findings of the non-zero code column metric.

## C Data Materials

### C.1 CinC2012 Data

CinC2012 Data (Silva et al., 2012) is a dataset proposed to predict the mortality of ICU patients in the CinC2012. It contains general descriptors such as age, gender, and ICU type and time series records like

heart rate, respiration rate, and serum glucose. In our experiments, we use the preprocessed version of this dataset from (Johnson et al., 2012), which is derived by applying simple extraction on the time-series features and excluding abnormal outliers in the physiological measurements. We then added the label of in-hospital mortality to the records, making 115 features in total. There are 4000 records for model training and another 4000 records for model testing, as split by the CinC authority. We use this dataset to evaluate the models' performance on mixed-type EHR data.

## C.2  PTB-ECG Data

PTB-ECG Data (Bousseljot et al., 1995) is a collection of ECG signals for heart disease diagnosis. We utilize a preprocessed version from (Kachuee et al., 2018) to carry on our experiments, where the signals are segmented and preprocessed from the original PTB Diagnostic ECG Database. The dataset contains 4046 normal patients and 10506 records with heartbeat classified as abnormal. Specifically, all the signals are cropped, downsampled, or padded to make each sample into a fixed dimension of 188. We use this dataset to explore models' ability to generate continuous medical time series data.

