# OpenReview forum: "EHRDiff : Exploring Realistic EHR Synthesis with Diffusion Models"
_TMLR — Accepted by TMLR_

### Review · Reviewer_F7A1 · 2023-11-21

**Summary Of Contributions:**

This paper is a resubmission based on the review comments provided earlier. The authors propose to synthesize EHR data using diffusion models. In particular they make the following contributions:
1. They apply diffusion methods from existing work (Song et al. and Karras et al.) to EHR data.
2. They perform experiments and compare the diffusion approach to GAN- based baselines, using diagnosis and procedure codes in MIMIC-III (a publicly available dataset). The utility and privacy metrics are adopted from previous work. The results show that while the performance of EHRDiff is comparable to existing work in terms of dimension-wise distribution, it has a significantly better result in terms of non-zero code columns, hence it overcomes the issue of mode collapse compared to GAN based methods. However, in terms of attribute inference risk and membership inference risk, GAN based methods perform better.
3. They also test diffusion on two datasets: CinC2012 and PTB-ECG, and show that EHRDiff performs best in terms of AUC.

**Audience:**

Yes

**Broader Impact Concerns:**

There are ethical implications associated with the work so I would recommend adding a Broader Impact statement. Previous work has highlighted issues of generative models and possible biases, which may also occur in EHR data. Hence, a discussion of similar concerns is necessary in the medical context.

**Claims And Evidence:**

Yes

**Requested Changes:**

Please see above.

**Strengths And Weaknesses:**

*Strengths*

Overall the paper is in a much better condition than its previous version. It is also generally well written and they've addressed all of my previous comments.

*Limitations*

The paper still requires further proofreading, e.g., missing full stops or rephrasing certain things like "achieves middling results".

---

### Review · Reviewer_bnZM · 2023-12-23

**Summary Of Contributions:**

The reviewers have fully addresses the previous comments.

**Audience:**

Yes

**Claims And Evidence:**

Yes

**Requested Changes:**

None

**Strengths And Weaknesses:**

I have no further comments to the current version.

---

### Review · Reviewer_r6Wj · 2024-01-02

**Summary Of Contributions:**

Due to privacy concerns, there is only limited access to large scale and high-quality EHR data. Therefore, it is of importance to synthesize high-quality EHR data. Previous methods focus on Generative Adversarial Networks (GAN) while this paper explores how to use the diffusion model to generate EHR data. This diffusion model-based method outperforms the previous methods and achieves the SoTA performance on this task.

**Audience:**

Yes

**Claims And Evidence:**

Yes

**Requested Changes:**

The authors have fully addressed the previous comments.

**Strengths And Weaknesses:**

Strengths:

To the best of the reviewer’s knowledge, this is the very first paper to apply diffusion models which often used for visual generation to EHR data.
The method is effective and generate more diverse data than previously proposed GAN based method.
The effectiveness of the method is validated on multiple datasets.
Weakness:

The method is a rather direct application of the current existing diffusion models. The proposed method did not update the original diffusion models designed for vision to the EHR data.

---

> ### Author Response · Authors · 2024-01-02
> **Responses to your comments**
>
> Thanks for your time reviewing our manuscript and your constructive comments.

---

### Decision · Action_Editor_XDMC · 2024-03-11

**Recommendation:** Accept as is

**Comment:**

This is a "technical" resubmission due to the author's misunderstanding of the procedure. It is now ready to be published.

**Audience:**

Yes, although it is on the fringes of the interests of TMLR readers.

**Claims And Evidence:**

Yes, there is no doubt about this between the reviewers.